# Design and Validation of a Low-Level Controller for Hierarchically Controlled Exoskeletons

**DOI:** 10.3390/s23021014

**Published:** 2023-01-16

**Authors:** Connor W. Herron, Zachary J. Fuge, Madeline Kogelis, Nicholas J. Tremaroli, Bhaben Kalita, Alexander Leonessa

**Affiliations:** Terrestrial Robotics Engineering and Controls (TREC) Laboratory, Virginia Tech, Blacksburg, VA 24060, USA

**Keywords:** hierarchically controlled system, series elastic actuators, sensors, microcontroller, communication

## Abstract

In this work, a generalized low-level controller is presented for sensor collection, motor input, and networking with a high-level controller. In hierarchically controlled exoskeletal systems, which utilize series elastic actuators (SEAs), the hardware for sensor collection and motor command is separated from the computationally expensive high-level controller algorithm. The low-level controller is a hardware device that must collect sensor feedback, condition and filter the measurements, send actuator inputs, and network with the high-level controller at a real-time rate. This research outlines the hardware of two printed circuit board (PCB) designs for collecting and conditioning sensor feedback from two SEA subsystems and an inertial measurement unit (IMU). The SEAs have a joint and motor encoder, motor current, and force sensor feedback that can be measured using the proposed generalized low-level controller presented in this work. In addition, the high and low-level networking approach is discussed in detail, with a full breakdown of the data storage within a communication frame during the run-time operation. The challenges of device synchronization and updates rates of high and low-level controllers are also discussed. Further, the low-level controller was validated using a pendulum test bed, complete with full sensor feedback, including IMU results for two open-loop scenarios. Moreover, this work can be extended to other hierarchically controlled robotic systems that utilize SEA subsystems, such as humanoid robots, assistive rehabilitation robots, training simulators, and robotic-assisted surgical devices. The hardware and software designs presented in this work are available open source to enable researchers with a direct solution for data acquisition and the control of low-level devices in a robotic system.

## 1. Introduction

Humans experience physical challenges in everyday life, where exoskeletons and other robotic systems offer assistance in rehabilitation and recovery, as well as an enhancement of normal human motor control capabilities. As per the World Health Organization, approximately one billion people, or 15% of the world’s population, experience some form of disability and are in need of an exoskeleton or assistive device [1]. In 2020, nearly 256,000 injuries to U.S. private industry workers were due to overexertion resulting in days away from work [2]. These injuries are often caused by repetitive lifting, lowering, pushing, and pulling heavy objects, most often leading to back impairment, constraining the overall mobility [3]. On another note, degenerative diseases and muscle weakness can have a substantial impact on a person’s day-to-day life, limiting one’s ability to perform daily activities without considerable physical support [4]. Elderly or impaired individuals experience difficulties in balance control while transitioning from sitting to standing, walking indoors and outdoors, and navigating stairs [5,6,7,8]. Overall, exoskeletons and other robotic devices offer independence to injured workers, elderly, and impaired or disabled individuals, who are limited in mobility and require additional care. These devices can be utilized in a variety of applications, including personal, industrial, military, and medical applications, to compensate for unfortunate physical disabilities or even enhance normal human capabilities [4,9].

Powered exoskeletons and other medical robotic devices require a complex network of sensors, actuators, and embedded devices communicating at real-time rates to achieve safe, robust control for rehabilitative or assistive behaviors. Exoskeletons are devices that physically connect to a human, augmenting the biological movement of the joints for training exercises, rehabilitation, or permanent support [10,11]. For example, the LOPES exoskeleton was designed for interactive gait rehabilitation using EMG measurements on the leg muscles and force sensory feedback to achieve several levels of autonomy for different training strategies. This exoskeleton utilizes Bowden-cable-driven series elastic actuators (SEAs) with a limited stiffness rate [12]. In the next generation design, the LOPES II exoskeleton achieves a gait-phase dependent stiffness, allowing for the spring rate to be increased to accommodate training in high-impedance mode for severely impaired patients [13]. In another work, the active leg exoskeleton (ALEX) uses an “assist-as-needed” approach for the neuromotor training of gait rehabilitation for patients with walking disabilities. The device uses sensor feedback from load cells, joint encoders, and force–torque sensors in its force-field controller (FFC) driven by linear actuators [14]. The MIRAD sit-to-stand assistive exoskeleton utilizes variable stiffness SEAs to assist in lower-body behaviors while delivering substantial compensation to the knee and hip joints [15]. In a different application, the Upper Body Sarcos exoskeleton was utilized for delivering upper body haptic feedback for virtual reality applications using whole-body control optimization approaches [16]. Finally, the RoboKnee is a one-degree-of-freedom exoskeleton driven using low-impedance SEAs, where instead of rehabilitating the user, it provides enhanced strength in the knees for carrying heavy loads [17].

These exoskeletal devices are driven using multiple configurations of SEAs for controlling each of the joints in the system [18,19,20,21,22]. SEAs are often driven using electric motors attached to a gear train with an elastic element. This elastic element is a passive spring that acts as a low-pass filter within the mechanical interface between motors and loads, resulting in more accurate and stable force control [18]. Variable stiffness actuators (VSAs) allow a control designer to mechanically vary the impedance level to improve the torque interaction between the robotic system and its surroundings depending on the task [23,24]. These devices commonly utilize disturbance or Luenberger observer approaches to maintain the robust force control of the actuator in the presence of nonlinearities or unmodeled dynamics [11,21,25]. Overall, SEA devices provide several mechanical advantages, including shock tolerance, more stable force control, and safer interactions between the robot and the human [18].

While exoskeletons provide direct contact on a human for rehabilitation and assistance, other medical devices with similar hardware can assist in a variety of other tasks. These medical devices include teleoperation for robotic surgery with haptic feedback [26], assistive rehabilitation [27], emotional and mental care [28], robot-assisted surgery with speech-based communication [29], and even training simulators for experienced physicians in assessing and evaluating ankle clonus, a set of involuntary and rhythmic muscle spasms [30]. Additionally, humanoid robots are commonly driven using similar sensor feedback and actuation approaches to exoskeletons and other medical devices. For example, the humanoid robots Valkyrie [21], Hume [31], SAFFIR [32], THOR [33], ESCHER [34], PANDORA [35], and DRACO [36] are driven using a variety of SEA designs with force control to achieve whole-body dynamic motion.

From the above-mentioned literature, it can be observed that there are limited works that specifically focus on the electrical and networking design of the low-level device within an SEA-driven robotic system. Exoskeletons, humanoids, and other medical robots often utilize similar sensors and hierarchical control strategies, where the development of the same hardware is unnecessarily time-consuming, expensive, and outside the scope of the core research objectives. Hence, the knowledge gap can be summarized as:An in-depth analysis and description of the low-level device and the required electrical designs for sensor acquisition, which may contain analog filters and other critical elements that dictate system dynamics and control performance.An overview of the networking strategy between the high-level PC and the low-level sensor acquisition devices, which typically have computation constraints, where the completion of communication, data acquisition, and control tasks is not guaranteed within a communication step.Results that validate the system performance in sensor acquisition and networking, separating the performance of the hardware from the rehabilitation strategy to better prepare the researcher to investigate the core research objectives of the robotic device and ensure the safety of the operator.A lack of readily available cost-effective, open-sourced, and flexible low-level data acquisition and control solutions for SEA-driven robotic systems.

The purpose of this work is to propose a low-level controller, which is a hardware component that collects sensor feedback and sends control commands to two separate linear series elastic actuators (LSEAs), communicates these data with an external high-level controller, and achieves internal critical real-time deadlines. This low-level controller is flexible in the sense that it can be a general solution to communicating and controlling multiple types of SEAs, whose sensor feedback and control input approaches are the same. These robotic systems often use control strategies that separate high and low-level controllers, which are stored on different hardware components. This low-level controller is designed to collect sensor feedback and execute control inputs while communicating at real-time rates with a high-level controller. Overall, this modular device is applicable to any robotic system that utilizes SEAs, including humanoid robots, upper and lower body exoskeletons, and other medical robots.

Figure 1 displays the abstract exoskeleton design that the proposed low-level controller (red) is applicable to. In general, exoskeletons are made up of three main components: a high-level controller (blue), a set of low-level controllers (red), and SEAs (green). It is important to recognize here that there is a single high-level controller stored in an external PC, and several low-level controllers and actuators depending on the number of powered joints on the exoskeleton. Each low-level controller can collect sensor feedback and control two separate SEAs.

The high-level controller contains the overall human–robot interaction control algorithm, where several approaches exist, including impedance control [12,14,15], admittance control [11,13,14], “force-field” control [14], and optimal control [11,16]. These control strategies are often paired with finite state machines to determine the intended motion of the user, and employ gait trajectory planners to achieve a whole-body behavior [11,13,14,15]. These algorithms are heavily dependent on the quality of the sensor feedback, utilizing encoders to measure joint and actuation positions [14,16,17], linear spring encoders [17,27], and force–torque sensors [12,13,14,16]. This high-level controller is often on board an external PC, where the computation capability is higher and completely dedicated to solving complex dynamic equations and optimization problems as quickly as possible [13,16]. Finally, the output of the high-level controller is a series of desired joint torque trajectories (blue arrow in Figure 1) that, if achieved, would execute the intended whole-body action depending on the algorithm.

The desired joint torque trajectories are sent to a low-level controller (LLC), which is a separate piece of hardware responsible for collecting sensor feedback and controlling two different LSEAs. The LLC is a combination of three main components: the EasyCAT Pro Shield [37], the Texas Instruments (TI) TM4C123GXL (TIVA) microcontroller launchpad [38], and the in-house-designed sensor interface shield. The EasyCAT Pro shield is an off-the-shelf networking device that communicates with the high-level controller using the EtherCAT (Ethernet for Control Automation Technology) protocol [39]. The TIVA microcontroller is an ARM processor with several debugging and design tools used to develop custom solutions for embedded systems [38]. Finally, the sensor interface shield is responsible for level-shifting the input motor commands (PWM) and conditioning the sensor feedback to be measured and comprehensive for the TIVA. In this work, the SEA subsystem uses a brushless DC motor connected to a gearbox and ballscrew mechanism through a reinforced belt, transmitting rotational energy from the motor into translational energy of the actuator [20,22]. The PWM command from the LLC is sent to a motor control unit, which is the combination of an off-the-shelf AMC AZBDC12A8 servo drive and an in-house-designed motor shield. The motor control unit sends the voltage switching commands to the multi-pole brushless motor to produce motion. As displayed in Figure 1, the LSEA (green) contains several different sources of sensor feedback: motor current, quadrature and absolute encoder, and force sensor feedback. The LSEA mechanism here utilizes an inline force sensor [20] instead of a linear encoder to measure elastic spring deflection [17,27]. Finally, a low-level controller can collect sensor feedback from a single inertial measurement unit (IMU) while communicating and controlling two LSEAs. Therefore, the main contributions of this work are highlighted as:An in-depth description of the electrical design and networking strategy for collecting sensor feedback and controlling SEAs while communicating with a high-level controller that can be generalizable to many exoskeletal and other medical robots;Two sets of open-loop results, which provide a validation of the expected performance of sensor feedback from a quadrature encoder, absolute encoder, motor current, force sensor, and IMU, along with ensuring proper networking with the high-level controller;The hardware and software presented in this work are available open source(https://gitlab.com/trec-lab, accessed on 18 December 2022), giving researchers a direct strategy for achieving the data acquisition and control of an SEA system;The developed open-source repositories along with the design details discussed in this paper also provide the flexibility to slightly modify or add additional sensors according to the particular applications.

In Section 2, the sensor interface shield and the internal circuit designs will be discussed. In Section 3, the motor shield and the IO channels, along with motor current conditioning, will be described in detail. In Section 4, the EasyCAT shield will be introduced, and the communication frame size and real-time networking challenges are discussed. In Section 5, a set of two open loop control results are provided, demonstrating the effective performance of the input command and sensing feedback. Lastly, in Section 6, final conclusions and future work are discussed.

## 2. Sensor Interface Shield

The LLC is made up of three components: the sensor interface shield, the Texas Instruments TM4C123GXL TIVA Launchpad, and the EasyCAT PRO Shield as shown in Figure 2. In our previous work [40], we introduced the design of the LLC, discussing the key design features and explaining and validating the analog filter designs in simulation. In this work, the full circuit designs are discussed in depth with results on hardware demonstrating the effectiveness of the networking, sensor feedback, and input command capability for controlling two linear series elastic actuators with one LLC.

As displayed in Figure 2, each LLC has a power input port with 10–15 V, a separate 5 V input, and a GND. The IO networking ports on the EasyCAT Pro Shield allow a low-level controller to network with the high-level controller at real-time rates using the EtherCAT communication protocol. On the sensor interface shield, there are four total feedback ports for force sensor, motor current, absolute encoder, and quadrature encoder feedback. These sensors are oriented as displayed in the linear series elastic actuator (LSEA) mechatronics portion of Figure 1. The sensor interface shield is separated into two sides, where two sets of IOs are available to control two sets of LSEAs.

Additionally, the LLC can collect feedback from a single inertial measurement unit (IMU), which can be directly wired into the header blocks of the sensor interface shield. Finally, three LEDs and oscilloscope pins for several key voltage feedback states are available for operational debugging purposes, giving a designer the ability to easily investigate an issue. The software architecture written for the TIVA microcontroller is discussed in [41], describing the distributed networking strategy and real-time operating system (RTOS). The focus of [41] is on determining the maximum LLC frequency while achieving sensor collection, networking with the high-level controller, computing the next control input, and sending the control input to the actuators. Additionally, on the sensor interface shield, a force sensor zeroing potentiometer (top of Figure 3a) is available to modify the calibration zero point, allowing an engineer to maintain a consistent positive and negative range of force sensing. The printed circuit board (PCB) layout of the sensor interface shield is displayed in Figure 3b, where the following subsections will focus on the underlying circuit design that allows for accurate sensor feedback collection and motor command.

### 2.1. Power Regulation

The logic power input of the LLC is critical, where a single, consistent 5 V signal is required to power the sensor interface shield, TIVA microcontroller, and EasyCAT Pro Shield. On the sensor interface shield, a majority of the circuits require a consistent 5 V power input for logic to achieve the expected hardware performance, where any drop in voltage could lead to improper or deficient results. On previous iterations of the sensor interface shield, a direct 5 V power input was utilized. Figure 1 depicts that several LLCs are utilized for an exoskeleton depending on the number of LSEAs required. It was found that, as more LLCs utilized the same 5 V power input in parallel, this voltage line began to drop. On a robotic system currently being developed with 12 LSEA units and 6 LLCs [35], the 5 V power signal was found to only be 4.11 V at the end of the voltage line. This voltage drop is concerning because any insufficient hardware performance could lead to the critical injury of an operator within an exoskeleton during operation. Therefore, it is imperative on the sensor interface shield to utilize a voltage regulator that allows for a flexible range in power voltage to maintain a consistent 5 V signal across an LLC, even when voltage fluctuations occur on the logic power input. The BA7805FP-E2 chip is utilized with an input voltage range of 7.5–25 V, outputting a 5 V signal with a maximum current draw of 500 mA [42]. On a bench test, the linear voltage regulator has been found to maintain a 4.95 V power input and hence will be able to fully provide power to the EasyCAT PRO Shield, TIVA microcontroller, and sensor interface shield circuits.

### 2.2. CAN Capability

A robust communication method for various devices is CANBus, which is designed to manage a network of complex devices similar to EtherCAT. CANBUS allows for many types of sensors to be added onto a CANH and CANL signal line. These two wires are twisted to protect against external interference and can be routed to any amount of sensors. To differentiate each sensor added to a CANBUS, each additional module should have a specific CANID that the MCU will reference when communicating. The circuits for CANBUS utilize the MCP2551 chip, a high-speed CAN transceiver [43]. This gives TIVA the ability to both send and receive external sensors on the CANBUS that also have a CAN transceiver. A digital signal is sent/received on the TIVA from the TX/RX pins, where the CANH/CANL pins utilize an analog signal. A CANBus circuit is featured on the sensor interface shield to employ a variety of additional sensors or devices, such as IMUs or force–torque sensors, utilized on similar exoskeletal systems, such as [12,13,14,16].

### 2.3. Force Sensing

For exoskeletons and many robotics systems, it is often desirable to utilize high-level control strategies that output joint torques to achieve a dynamic behavior. Therefore, for the LSEA system shown in Figure 1, a linear load cell was utilized to measure the output force of the actuator applied onto an exoskeleton’s joint. For the LSEA subsystem in Figure 1, a linear load cell with a full Wheatstone bridge type III strain configuration was utilized to measure axial loads and reject bending loads. This type of load cell ensures that the measured forces are linear, where the actuator is designed to apply axial loads. Bending loads can occur from mechanical inaccuracies in the design and would certainly affect the performance because the force sensor and ball screw mechanism in the SEA are weak in the directions perpendicular to the applied [20]. Therefore, as displayed in Figure 4, the difference between the $V_{+}$ and $V_{-}$ is a good representation of the linear load even in the presence of small unintended bending loads. Due to voltage limitations on the LLCs, an excitation signal of 2.5 V was utilized, which also lowers noise caused by heat generation across the bridge resistors. $V_{+}$ and $V_{-}$ are differential signals on the mV scale representing a tension or compression load across the load cell. In order to measure the voltage signal using the TIVA’s 12 bit analog-to-digital converter (ADC), an instrumentation amplifier was utilized to scale the voltage signal to the 0–3.3 V voltage range. Shah [44] investigated force sensor amplification using the INA125 instrumentation amplifier, which provides an accurate measurement with a low single voltage supply range at +5 V.

The INA125 chip allows for a wide range of voltage gains from 4 to 10,000. For our applications, the LCM200 load cell by Futek was utilized with a rated output of 2 mV/V [45]. The rated output, $V_{RO}$, represents the maximum differential voltage reading by the $V_{+}$ and $V_{-}$ signals for a 1 V excitation voltage, $V_{exc}$. The required gain, *G*, can be calculated using the following equation:(1)$$V_{o}=G\cdot V_{exc}\cdot V_{RO}$$
where $V_{RO}=2$ mV/V at a maximum load with $V_{exc}=1$ V. The $V_{exc}=2.5$ V was utilized for our system because it lowers the noise from the load cell and is the only available option for the +5 V power supply. The output voltage, $V_{o}$, refers to the desirable voltage signal range of the output signal. In this case, the TIVA’s ADC voltage range is between 0–3.3 V. Therefore, it is desirable to have a full compression and tension range within 0–3.3 V, where the no-load point is at 1.65 V, meaning $V_{o}=3.3-1.65=1.65$ V. Using (Equation 1), the required gain can be found: G=330. Using the following equation provided in the INA125 datasheet [46], we can solve for the $R_{G}$:(2)$$G=4+\frac{60\;k\Omega }{R_{G}}.$$

For this application, $R_{G}=184.0\;\Omega$, where $R_{G}=R_{21}=180.0\;\Omega$ is displayed in Figure 4. Finally, a key feature of the INA125 chip is a voltage offset capability, which creates a “psuedoground". The “psuedoground” capability allows an engineer to easily shift the no-load voltage without external components. In this case, it is required to maintain the full voltage signal within the 0–3.3 V range and avoid negative voltage inputs into the TIVA’s ADC. Using this feature, the no-load condition can be modified from 0 V to 1.65 V to give an equal measurable range for tension and compression. Therefore, only an additional potentiometer, $R_{7}$, is necessary as a simple voltage divider to drop the $V_{exc}=2.5$ V supply signal to $V_{IA}=1.65$ V to modify the “no-load” voltage. In this case, the “no-load” voltage refers to zero tension or compression on the force sensor and is not to be confused with a voltage output load. A potentiometer is convenient here, where each force sensor has a slightly different “no load” voltage, allowing for adjustment during the calibration of the sensor. This gain setting and “no-load” voltage adjustment ensures a proper voltage output of the full tension and compression range of the load cell to the full range of the TIVA’s ADC. Finally, a second-order low-pass Sallen–Key filter was utilized to remove high frequency noise and eliminate signal aliasing with a cut-off frequency, $f_{c}=482$ Hz. This circuit design was presented in prior work [40].

### 2.4. Motor Input

The advanced motion control (AMC) AZBDC12A8 motor controller is an off-the-shelf servo drive for brushed and brushless DC motors [47]. It allows for supply voltages from 20–80 VDC with a peak output current, $I_{peak}$, of 12 A and a continuous current of 6 A with internal protection to prevent over-voltage, over-current, over-heating, and short circuits. A PWM duty cycle dictates the output motor current and a digital direction pin dictates the direction of motor rotation. The PWM duty cycle is related to the output motor current, $I_{o}$, by the following equation:(3)$$I_{o}=I_{peak}\cdot \frac{PWM}{100}.$$
After careful testing, it was found that the PWM and direction voltage logic level must be 5 V logic to ensure a proper performance. The TIVA microcontroller only outputs a 3.3 V logic signal; therefore, an additional step-up circuit was necessary to convert 3.3 to 5 V logic. An important constraint of the step-up circuit is that it must provide little distortion to digital signals at 10–25 kHz signals. Previous circuit designs added slight distortion to the PWM signal, converting a square wave to a more triangular wave. At lower PWM values such as 3% or lower, the PWM signal was no longer measured by the motor controller, creating an input deadzone. Therefore, as displayed in Figure 5, the high-speed CD40109BPW MOSFET chip was utilized for quickly level shifting the square wave signals, with little effect on the voltage shape at 20 kHz as previously shown in [40]. This chip has four IO ports to convert two sets of PWM and direction voltage signals for commanding the two motors that drive their separate LSEA systems.

### 2.5. Encoders

In order to control each joint of the exoskeleton, the positions of the joints and motors must be precisely measured. This is accomplished using absolute and relative or incremental encoders. Absolute encoders utilize optical discs or magnetic fields to indicate the a digital position of a shaft relative to its mounted frame from startup. Therefore, the position of a joint can be immediately determined from the initial measurement. On the other hand, incremental encoders determine relative positions with respect to startup. Incremental encoders are often an additional package for motors that can provide a higher resolution of the position and estimated velocity on the applied joint of the exoskeleton. However, an incremental encoder often requires a calibration phase on startup and is more susceptible to measurement nonlinearities such as backlash or belt slipping when used for estimating a joint position and velocity. Improvements in joint measurement accuracy could likely be achieved using both absolute and incremental encoders with a Kalman filter, which is a common choice for sensor fusion applications involving encoders [48,49].

The absolute encoders communicate using SSI, where the LTC490 differential transceiver converts the differential clock and data signals from the sensors to signal-ended for the SSI module of the TIVA microcontroller. The LTC490 is a differential driver and receiver pair, and is a differential bus for data transmission. The driver pins are used to create an inverted clock signal that synchronizes data transfer between the encoder and microcontroller. As displayed in Figure 6, the PCB connects the driver pin (pin 3) to the clock for the SSI module on the TIVA. The inverted and non-inverted clock signals on the encoder can then be connected to the driver outputs: pins 6 and 5, respectively. The data transmission signals from the encoder are also differential. The receiver function of the LTC490 chip compares the two signals and outputs the non-inverted input signal, reducing noise and error in the reading.

A quadrature encoder is a type of incremental encoder that utilizes two channels (often denoted as channels *A* and *B*) to indicate a relative position change. As displayed in Figure 7a, sensors often provide three signals and their inverted pair: channels *A*, *B*, and *i*. Channels *A* and *B* are two square signals 90 degrees out-of-phase, where the high–low relationship indicates a rotation direction. The resolution of a quadrature encoder is signified by a number of counts per turn, where a count in a clockwise or counterclockwise direction signifies a rotation of the shaft as measured using the channel *A* and *B* signals. This measurement can provide a relative change in position from the startup location. Channel *i* refers to the index, which represents a full 360 degree rotation. In this application, the index channel was not utilized, but may prove useful for calibrating the sensor after an unexpected power cycle. The inverted signals are unnecessary for measuring the motor position, but provide an additional level of redundancy in a measurement. Unlike an absolute encoder where digital signals indicate an exact position, a relative encoder is more prone to error, where a single misinterpreted count measurement will make the total position wrong from that point on. Therefore, it is extremely important to utilize the redundant inverted signals to lower the number of errors. In previous designs of the sensor interface shield, the inverted signals were ignored and approximately 50% of the tested quadrature encoders had errors in measuring the motor position and estimating the motor velocity. An easy way to determine if the relative position is interpreted correctly is to rotate the shaft 10 times clockwise and then 10 times counterclockwise. If the position does not return to the initial value, then there is a mistake in the measurement. As displayed in Figure 7b, the SN75175 differential line receiver can be utilized to compare channel *A* and $\bar{A}$ to provide a higher accuracy representation of the channel *A* signal in the presence of noise. Both channels *A* and *B* utilize the comparator circuit, where a single SN75175 chip provides four separate differential line receiver circuits, providing enough ports for two motors. Finally, the MC3486 and AM26LS32 chips are denoted by the sensor manufacturer as additional recommended options of quadruple differential line receivers [50].

### 2.6. Inertial Measurement Unit (IMU)

Finally, the LLC is designed to communicate with a single inertial measurement unit (IMU) device. IMUs are commonly utilized for estimating the floating base of humanoid robots, exoskeletons, or drones within the world frame. Furthermore, multi-DOF robotic systems can employ a network of IMUs to aid in and improve the accuracy of joint state measurements when combined with encoders through sensor fusion [51]. As discussed in [35], complex robotic systems such as exoskeletons and humanoid robots often contain many nonlinearities, such as joint backlash, viscous friction, and unknown or nonlinear structural elasticity properties. These nonlinearities could have a substantial impact on dynamic control, leading to instability. Therefore, an IMU is a useful sensor to utilize for a variety of control applications in robotics.

The IMU used in this implementation is the MPU9250. The MPU9250 comes equipped with a three-axis accelerometer, three-axis gyro, and three-axis magnetometer [52]. There is a calibration process that is programmed into the low-level code that zeros the biases for the gyroscope and the magnetometer. Communication to the IMU can be established via either I2C or SSI. For this implementation, I2C is used to communicate with the MPU9250. Once data are read via the communication protocol, the biases are applied from the calibration process to achieve the accurate value needed. The data are then added to the EtherCAT frame and sent to the high-level controller to estimate the global position and orientation of the robot using the Madgwick sensor fusion algorithm [53].

## 3. Motor Shield

The motor shield is an in-house-designed shield that sits on top of an off-the-shield AMC AZBDC12A8 servo drive to create the motor control unit package. As displayed in Figure 8, the motor control unit is utilized for driving the brushless motors of an LSEA. As discussed in Section 2.4, a PWM and high/low direction signal are utilized to indicate a desired output motor current. The motor controller utilizes Hall sensor feedback from the brushless DC motor with its own internal control loop to power the poles to achieve the desired output current. Additionally, the motor controller provides motor current feedback as a bipolar analog signal. The motor shield has four ports: power source input, motor power output, Hall sensor feedback, and an IO port for the PWM and direction input and motor current output. A layout of the motor shield is displayed in Figure 9, where the motor current conditioning and voltage rail circuits can be observed.

### 3.1. Motor Current Feedback

The AZBDC12A8 motor controller provides motor current feedback as a 4 A/V bipolar analog signal, where a peak output of 12 A translates to a maximum ±3 V wave linearly representing the motor current [47]. However, the motors used for the LSEA systems have a maximum continuous current limit of 3.68 A. Therefore, the motor current voltage feedback range is limited to 4 A, providing more accurate sensor data for the operational current range. The motor current voltage reference can be measured using the TIVA’s ADC, where the ADC voltage range is 0–3.3 V. Therefore, a conditioning circuit is necessary to bias and scale the ±1 V wave of the motor current reference to the ADC voltage range. As displayed in Figure 10, the conditioning circuit is made up of four components: the passive RC and active twin-T notch filters, the gain, the bias circuit, and the summing amplifier. The two-stage filter is a combination of an active twin-T notch filter and a passive low-pass RC filter to remove 60 Hz noise from power supplies and high-frequency noise, $f_{c}=397.9$ Hz, respectively. High-frequency noise has been observed at 31 kHz, and is caused by the motor controller’s switching frequency, which has a dramatic impact on motor current feedback noise and is completely removed by the filter stage of the conditioning circuit.

As shown in Figure 10, the active twin-T notch filter features two operational amplifiers, allowing for a flexible range of quality values, *Q*, which signify the shape of the notch. In the configuration here, the rejection frequency, $F_{o}=60.0$ Hz, is chosen to remove the AC noise caused by the power supply, which has significant effects on the performance. Based on the chosen design configuration of Figure 10 when $R=R_{1}=R_{2}=2R_{3}$ and $C=C_{1}=C_{2}=\frac{1}{2}C_{11}$, the abbreviated transfer function of the active twin-T notch filter can be written as
(4)$$V_{NOTCH}\left(s\right)=\frac{s^{2}+{\left(\frac{1}{RC}\right)}^{2}}{s^{2}+\left(\frac{1}{RC}\right)\left(\frac{4R_{4}}{R_{4}+R_{5}}\right)s+{\left(\frac{1}{RC}\right)}^{2}}V_{S04}\left(s\right),$$
where $V_{S04}\left(s\right)$ refers to the raw motor current voltage signal from the AMC servo drive. The ratio of $R_{4}$ and $R_{5}$ decides the quality factor, $Q=(1+R_{5}/R_{4})/4=1.0$, and the choice of *R* and *C* decides the attenuated frequency, $F_{o}$, where $F_{o}=1/\left(2\pi RC\right)=60.0$ Hz for our application [54]. The RC low-pass filter is simply represented by the following transfer function:(5)$$V_{filtered}\left(s\right)=\frac{1}{R_{6}C_{13}s+1}V_{NOTCH}\left(s\right)$$
where this first-order, unity gain, passive filter has a cut-off frequency of $f_{c}=397.9$ Hz for $C_{13}=100\;\mathrm{nF}$ and $R_{6}=4\;k$Ω. This two-stage filter is denoted as NOTCH_RC_FILTER in Figure 10. The filtered voltage signal, $V_{filtered}$, is still a bipolar signal of ± 1 V that must be amplified and biased to 0–3.3 V at the conditioning output, $V_{out}$, to be measurable by the TIVA’s ADC. This process can be summarized by the following equation:(6)$$V_{out}=G\cdot V_{filtered}+V_{BIAS}$$
Therefore, a gain, G=3.3/2=1.65, is required, where $V_{out}$ would be a ±1.65 V signal when $V_{BIAS}=0$ V. Thus, finally, $V_{BIAS}=1.65$ V to fully capture the filtered motor current feedback within the ADC voltage range. A simple non-inverting active gain circuit was used to achieve G=1.65 from the $V_{filtered}$ signal represented by the following equation:(7)$$G=1+\frac{R_{8}}{R_{7}}$$
where $R_{8}=6.5\;k$Ω and $R_{7}=10\;k$Ω. A voltage divider is represented by the following equation:(8)$$V_{BIAS}=\frac{R_{14}}{R_{13}+R_{14}}V_{+}$$
where, if $V_{+}=5$ V, and the readily available resistors are chosen to be $R_{13}=2\;k$Ω and $R_{14}=1\;k$Ω, this results in $V_{BIAS}=1.67$ V, which is a sufficient offset for this application. Finally, this bias voltage is passed through a buffer to maintain isolation between the bias and summing amplifier circuits impedance. The voltage divider is displayed in the BIAS section of Figure 10. Finally, the summing amplifier adds the $V_{BIAS}$ and $G\cdot V_{filtered}$ signals to create $V_{out}$, as previously shown in (Equation 6). The summing amplifier can be represented by the following equation:(9)$$V_{out}=\left(\frac{R_{12}}{R_{11}}+1\right)\cdot \frac{R_{10}G\cdot V_{filtered}+R_{9}V_{BIAS}}{R_{9}+R_{10}}$$
where it can be seen that, if $R_{9}=R_{10}=R_{11}=R_{12}=1\;k$Ω, then $V_{out}=G\cdot V_{filtered}+V_{BIAS}$.

### 3.2. Voltage Rails

As the current sensing filter works to provide a clean signal within the microcontroller’s ADC range, the voltage swing at different points in the circuit can vary between approximately −1.6 V and 3.3 V. The swing requires voltage rails that encompass the full range of voltage to prevent saturation of the operational amplifiers. The low-level controller provides a 5 V power signal, $V_{IN}$ in Figure 11, at the IO port, but the ideal rail voltages are at least equal to the range of voltages seen by the current sensing circuit.

To solve this issue, a split-voltage power supply was designed using an LTC3388 chip [55]. This high-efficiency step-down regulator, also known as a buck converter, works by charging and discharging an inductor as a switch, which causes the circuit to be open and closed to the input voltage. To achieve the split-voltage power supply, one chip is used in a standard buck topology and a second with the output connected to the ground as shown in Figure 11. This creates a negative potential on the GND pin. The result with an input of 6 V to 12 V is a regulated ± 5 V supply (denoted as $V_{+}$ and $V_{-}$ in Figure 11). Using the 5 V supply, $V_{IN}$, from the microcontroller, the split-voltage supply can provide a positive and negative voltage between 4.5 V and 5 V of magnitude, which is sufficient for the range seen by the current sensing circuit at various nodes.

## 4. High-Level Networking

The EasyCAT Pro is a simple EtherCAT slave interface based on the LAN9252 chip, which is used to help establish communication from a high-level computer system to low-level controllers. Displayed in Figure 12, the EasyCAT Pro essentially allows any embedded system within a distributed microcontroller system to function as an EtherCAT slave. The board directly communicates with embedded systems using the SSI communication protocol. Depending on the amount of data packaged in a communication frame, the EasyCAT Pro must be flashed to a particular byte-sized array (default options are 16, 32, 64, and 128 bytes) to fulfill that data storage. Once the board is flashed, proper interfaces must be designed for the high and low-level devices to package the EtherCAT frames within the correct data size. Data transfer between a master (high-level) and slave (low-level) is treated as a “hand-off” system, meaning that the embedded device reads and publishes new data at the same time.

There is one major challenge present with regard to synchronization when working with EtherCAT. A major issue arises when the master computer executes the communication loop faster than the embedded system can process. If the communication loops are not properly managed at the high and low-level, EtherCAT frames can be skipped, where the update rate for each of these devices is unknown. If the update rate of the two sets of devices is unknown, then the exact timing of sensor collection is not guaranteed and copies of previous communication frames may be received. In prior testing with a master and slave communicating at update rates of expected 1000 Hz, it was found that the slave did not use a fast enough bit rate for SSI between the EasyCAT Shield and TIVA. Instead of a 1000 Hz update rate, the actual update rate of the low-level device was 40 Hz. In our prior work [41], this problem was solved by the creation of a master process ID (MPID). The MPID refers to the update frame value set by the master when communicating with a low-level device. The low-level device packages the same MPID as a part of its sensor feedback frame to signal the completion of the associated task. Therefore, by monitoring the MPIDs, the master is capable of understanding and managing the synchronization of each embedded system in the distributed system. By simply looking at how long the MPID takes to update, one can have a general understanding of how long the embedded system takes to complete one communication/control cycle.

There are various signals that can be sent to the slave (low-level controller) from the master (high-level controller) and vice versa (denoted as “Signal-From-Master” and “Signal-To-Master” from the low-level controller’s perspective). These signals command a mode of operation, which can be seen below. The most common signal to send is the CONTROL signal, which commands the controllers to achieve a desired actuator torque output. The LLC populates the “Signal-To-Master” value with a HALT signal only if an error occurs. Such an error could involve triggering the virtual emergency stop or sensor irregularities. Otherwise, if there are no issues, the low-level controller simply echos the signal that the master has sent to it.

**HALT:** Triggers the LLC’s emergency-stop and turns motors off.**CONTROL:** Commands the LLC to achieve a desired actuator torque output. The LLC then sends the master the most updated sensor values.**IDLE:** Puts the low-level controllers in standby.**INITIALIZATION:** Sends initialization data that the microcontroller needs before it can start.

During a CONTROL signal, the LLC sends all of the sensor feedback from the two LSEAs, including the IMU. The full data signal along with the corresponding types and byte sizes can be seen in Table 1. Often for exoskeletal systems, force–torque sensors are utilized for measured ground reaction forces and torques between the robotic device and the environment [12,13,14,16,33,34]. Thus, 24 bytes in the EtherCAT frame are allocated to force–torque data collection. Finally, an empty “Remaining Bytes” fills the rest of the EtherCAT frame with zeros to maintain the correct frame size. In Table 2, the master sends direct PWM commands along with the motor’s spin direction. In the future, this CONTROL signal will likely change to commanded joint torque setpoints, where the low-level controller will feature an onboard control loop running at a faster rate.

The high-level controller is developed around the IHMC Open Robotics Software (ORS) repository, which contains several software packages for robotics applications in simulation or runtime [56]. IHMC’s ORS contains packages for motion capture, whole-body control, networking, state estimation, path planning, and sensor processing focused on legged robots, but applicable to a variety of robot systems. For EtherCAT communication, an additional repository is required that uses a Java wrapper around the Simple Open EtherCAT Master (SOEM) software [57]. There are APIs built within IHMC for various different EtherCAT slaves. EasyCAT and EasyCAT Pro are both supported by the API.

## 5. Results and Discussion

To validate the performance of the low-level controller (LLC), a pendulum test bed was designed and outfitted to measure all sensors for a single LSEA system, including the IMU’s measurements. The pendulum is made almost entirely of aluminum components, has a 0.3 m level arm, and a 10 kg weight at its end. As discussed in [18,21,25], series elastic actuators are often placed on pendulum test beds for measuring input–output relationships for modeling purposes. In [21,25], the input motor current and output force measurements were utilized to design disturbance-observer-based controllers, which help to remove the nonlinear dynamical effects of joint and ball screw backlash, viscous friction, and additional unmodeled dynamics. In the future, this system will serve as a contained environment for various controllers and modeling approaches.

As displayed in Figure 13, the high-level controller sends direct PWM commands in the form of % float type values, which the LLC interprets and converts into proper digital waveforms transmitted to the motor control unit (MCU). As discussed in Section 2.4 and Section 3.1, the MCU interprets the PWM signal as a desired output motor current and, using its own internal control loop, will achieve that desired trajectory. The measured motor current feedback (mA), $i_{m}$, is conditioned on the MCU, and then measured by the LLC microcontroller’s ADC module. The quadrature encoder measures the motor position (rad), $q_{m}$, relative to initialization. As discussed in Section 2.5, an absolute encoder measures the absolute position of the pendulum joint, where the calibrated zero position is located at the bottom of the pendulum’s swing. The pendulum joint angle (rad), $q_{j}$, is positive when the pendulum swings to the left, which can be observed in Figure 13. The force sensor measures the reaction of the actuated output force applied on the joint, $f_{o}$. For control purposes and in the results, this force vector was flipped to measure the actuator’s output force, $f_{a}$. Finally, according to Figure 13, the IMU is attached at the top of the weight, where the measured accelerations, angular velocities (gyroscope), and magnetic field intensity (magnetometer) can be collected. The coordinate axes displayed in Figure 13 refer to the acceleration and angular velocity directions, while the magnetometer measures with respect to a different coordinate system according to [52].

All of the sensor feedback is collected within the LLC at 1000 Hz and communicated to the high-level controller at a consistent 400 Hz. Additionally internal to the LLC is a virtual emergency stop that immediately sends a zero PWM command if a measured joint angle or force sensor value is too high. This additional form of security was crucial in the development process to ensure the safety of the hardware during testing and validation. For validating the performance of the LLC, two sets of open-loop PWM commands were sent from the high-level controller to the LLC: a 1 Hz, 1000 mA amplitude sinusoidal input, and a step-wise input of 250 mA, 500 mA, 750 mA, and 1000 mA at 0.5 Hz.

As displayed in Figure 14 and Figure 15, the commanded PWM (%), motor current (mA), joint position (rad), motor position (rad), actuator force (N), acceleration (m/s^2^), angular velocity (°/s), normalized magnetic field intensity (μT), and master process ID number are presented for the full LSEA sensor feedback, including the IMU. In the first case, an open-loop 1 Hz, 1000 mA amplitude sinusoidal input was sent from the high-level controller to the low-level controller as a PWM (%) signal in floating point representation, as displayed in Table 2. As discussed in Section 2.4 and Section 3.1, the PWM is directly related to a motor current output value that the MCU’s internal control loop achieves. Therefore, in Figure 14, the motor current (mA) is directly in phase and achieves a 1000 mA signal with a small amount of noise. While the motor current feedback represents the output motor current, there are scenarios where the PWM and the output motor current are not exactly the same. Based on the motor current conditioning circuit in Figure 10, the gain was chosen to maximize the ADC range for a ±4 A signal since the brushless DC motor has a maximum continuous current limit of 3.68 A. However, there might be instances where the desired motor current is above 4 A. In this case, the motor current would display a saturated value of 4 A. However, it is useful to maximize the ADC resolution within the most central operating range, and this is dependent on control requirements. Therefore, this gain might be lowered below 1 to ensure the full 12 A of motor current depending on the controller performance in the future. Finally, the motor current feedback contains a 0.1 A offset, likely caused by resistor and capacitor tolerances in the motor conditioning circuit, which is easily removed in software.

From Figure 13, it can be observed that the motor position (rad) and joint position (rad) are directly related, where the motor utilizes a reinforced belt to drive a ball-nut–ball-screw assembly [20]. Therefore, in Figure 14, the joint position and motor position are directly related, where the larger gain in the motor position is attributed to the gear ratio of the arrangement. The joint position has a slight offset from the zero position, which is caused by the test starting close to, but not exactly, at zero, whereas the motor position is relative and always starts at zero. While these sensors display a similar sensor feedback performance, there are different scenarios where each sensor is useful. The joint encoder is a simple and accurate approach to directly measuring a joint angle. However, the motor encoder can display the same data at a finer resolution, where a single motor rotation likely produces an immeasurable state change in the joint encoder. However, using the motor encoder in forward-based kinematics approaches does not consider unmodeled dynamics such as backlash or belt slipping. Overall, the joint position and motor position display a very clean performance, where both sensors will be effective for measuring the state change depending on the controller approach.

The actuator force feedback (N) is very clean, where the signal displayed in Figure 14 is the input motor current, and the output actuator force likely has fourth or fifth order dynamics. A high-impedance (pendulum arm does not swing) LSEA system is often modeled as a second-order mass, spring, and damper system [18,21,24,25]. However, based on initial testing, this LSEA system seems to indicate a third-order dynamic relationship. When the pendulum is released, an additional set of second-order dynamics can be found for the full fourth or fifth-order relationship. Further, the modeling and controller development will be investigated in future work. Lastly, it has been found that, after a few minutes of testing, noise may be induced on the force feedback caused by the heat generation of the voltage regulator depending on the voltage level of the input. Using the alternative 5 V input removes this noise, and the location of the voltage regulator will be a focus in future work.

The inertial measurement unit (IMU) provides sensor feedback for the acceleration (m/s^2^), angular velocity (°/s), and magnetic field intensity (μT). The acceleration measurements indicate a proper operation, where, based on the sensor orientation give in Figure 13, a majority of the acceleration vector is in the negative *x*-direction due to gravity. Acceleration due to gravity is measured in the opposite direction from a physics convention, where the IMU is held stationary in the presence of a gravitational field. In short, accelerating upwards is measured the same as gravity, which points downwards. For the angular velocity (°/s), the coordinate frame matches the acceleration reference, where a majority of the motion is in the *z*-direction. Therefore, these results match the expected performance based on the sinusoidal input shape. The magnetic field intensity (μT) is normalized in order to display the *x*, *y*, and *z* direction performance matches with offsets of $H_{x}=-65.5\;\mu$T, $H_{y}=-217.0\;\mu$T, and $H_{z}=125\;\mu$T. The coordinate system is different for the magnetometer based on the datasheet [52]. Overall, the magnetometer readings match the expected sinusoidal response once normalized and no drifting occurs in the measurement because of the placement of the sensor and the input amplitude.

Finally, the master process ID (MPID) value is an incrementing value representing the communication loop between the high and low-level system [41]. This value increments to 200 so that it can be represented as an 8 byte value in the EtherCAT Frame. In Figure 14, the MPID does not start at 0 when the time is equal to zero because communication starts before the test begins to ensure a proper performance of the system before enabling power. The triangular wave shape in time indicates that the high and low-level systems are able to communicate at a consistent rate of 400 Hz. If communication had stalled or slowed down, the MPID shape would no longer have a constant slope and would distort slightly. As discussed in [41], robotic control is heavily dependent on the communication, where improvements that maximize the networking rate will be addressed in future work.

In the other case displayed in Figure 15, an open-loop step-wise input of 250 mA, 500 mA, 750 mA, and 1000 mA at 0.5 Hz is sent from the high-level controller to the low-level controller. Similar to the first set of results, the PWM and motor current display a direct relationship. The motor current has a small amount of noise, but overall produces the expected step-wise commands with a small amount of transient time between set points. Additionally, the motor current feedback has the same 0.1 A offset, which was removed in software and is not a concern.

The joint and motor positions (rad) display an expected consistent relationship that matches the input command with a long transient period between setpoints. Based on Figure 15, a 1 s transient period occurs before reaching a steady state. The joint and motor position do not return back to zero within the 8–10 s period, but this is due to the amount of viscous friction or stiction in the ball screw, which slows the pendulum arm from returning back to the zero position. Even if the time horizon were increased in this test, the pendulum would not reach the zero position and instead would be held on a slight offset caused by this friction.

The actuator force (N) for the step-wise inputs display a more intuitive response in Figure 15 when compared to the sinusoidal results. The actuator force increases as the input increases, where a small sinusoidal waveform can be seen to be caused by the multi-order dynamics of the LSEA and pendulum system. Similar to the joint and motor position, the stiction in the ball screw slows the pendulum when the PWM is set to 0 at a time of 8–10 s. Since this measurement represents the applied force onto the pendulum joint, the frictional forces caused by stiction are measured by the sensor. Therefore, similar to the joint and motor encoder, the actuator force would settle to a non-zero value because of the friction.

The acceleration has a slight increase in the *y* and *z* components, where the majority are still in the *x*-direction caused by gravity. The *y* and *z* acceleration increase is likely caused by the sensor’s harness coming loose during testing, and does not pose a potential issue in the future. A similar set of results can be seen in the angular acceleration, where sudden movement can be seen in the *x*, *y*, and *z* components at a new set point. A majority of the motion is measured in the *z* direction as expected, but the *x* and *y* components display a bit of additional movement that is likely caused by the loose harness. Finally, the normalized magnetometer had a different set of offsets from the sinusoidal test, with offsets of $H_{x}=-5.6\;\mu$T, $H_{y}=-177.0\;\mu$T, and $H_{z}=31.4\;\mu$T, once again caused by the loose harness. However, a majority of the motion is indicated in the x-direction, which matches the expected shape of the input based on the orientation of the sensor [52].

Finally, the MPID indicates consistent communication between the high and low-level controllers at 400 Hz. A distortion to the slope would indicate that the communication has slowed, but Figure 15 has a constant slope for MPID. The MPID does not start at zero when time is zero because communication has already begun before the test starts.

## 6. Conclusions

In this work, a generalized low-level controller was developed for sensor collection, motor input, and networking with a high-level controller. A full overview of an abstract exoskeleton design is presented with a hierarchically controlled framework, which separates sensor collection and actuator output commands from computationally intense high-level algorithms to different hardware. A developed generalized low-level controller was designed to communicate and control two separate SEAs and collect data from an IMU. The internal circuit designs for the sensor interface shield and motor shield are presented for collecting and conditioning sensor feedback. The EtherCAT networking approach between the high and low-level controllers was discussed, with additional details on the challenges of device synchronization and update rates. The low-level controller was validated using a pendulum test bed driven using an SEA complete with full sensor feedback and IMU. The high-level controller sends two sets of open-loop input commands: a 1 Hz, 1000 mA sinusoidal input, and a step-wise input at 250 mA, 500 mA, 750 mA, and 1000 mA at 0.5 Hz. In both scenarios, all sensor feedback matches the expected performance, with minimal noise and irregularity. The multi-order dynamic response between the input motor current command and the output force can be observed, which is an expected and important relationship for effective force control. Overall, the performance of the developed low-level controller is highly effective for hierarchically controlled exoskeletal systems, which require low signal-to-noise ratios for safe control action in critical circumstances. The hardware and software discussed in this work are available open source (https://gitlab.com/trec-lab, accessed on 18 December 2022) to enable researchers with a direct strategy for achieving the data acquisition and control of an SEA system. In future work, the low-level controller will converge into a single PCB device to lower the overall height, and the joint torque controller will be developed using a disturbance-observer-based approach.

## Figures and Tables

**Figure 1 sensors-23-01014-f001:**
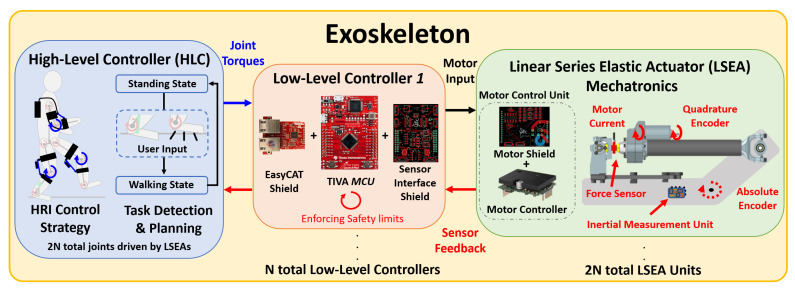
This represents the abstract exoskeleton design robotic system that we are designing the low-level controller for.

**Figure 2 sensors-23-01014-f002:**
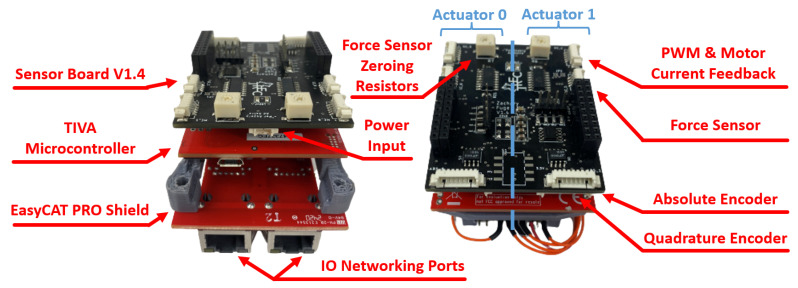
Port layout of low-level controller (LLC) made up of the sensor interface shield, the TIVA microcontroller, and the EasyCAT PRO Shield capable of networking with a high-level controller and controlling two LSEAs.

**Figure 3 sensors-23-01014-f003:**
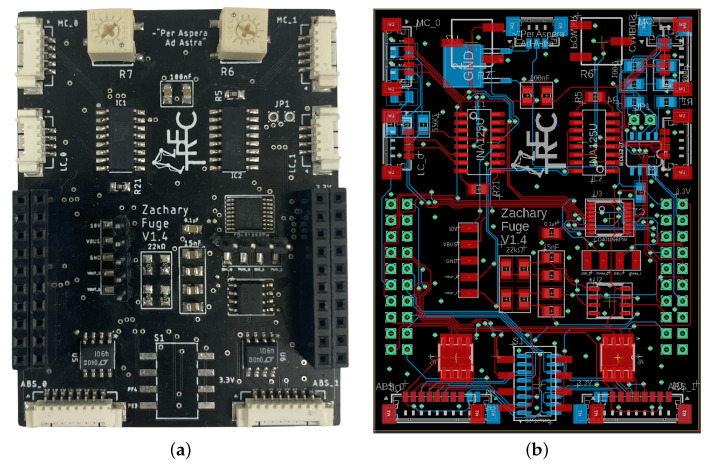
The sensor interface shield handles two sets of feedback for LSEAs systems, including force sensor conditioning, PWM logic step-up, quadrature encoder signal redundancy, and absolute encoder feedback. (**a**) Physical build of sensor interface shield. (**b**) PCB layout of sensor interface shield.

**Figure 4 sensors-23-01014-f004:**
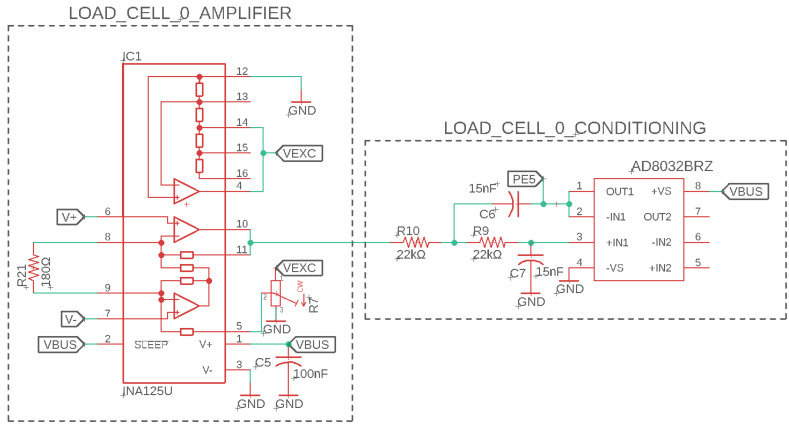
The INA125 instrumentation amplifier was utilized with a second order low−pass Sallen−Key filter to collect linear load cell force feedback with a Wheatstone bridge topology.

**Figure 5 sensors-23-01014-f005:**
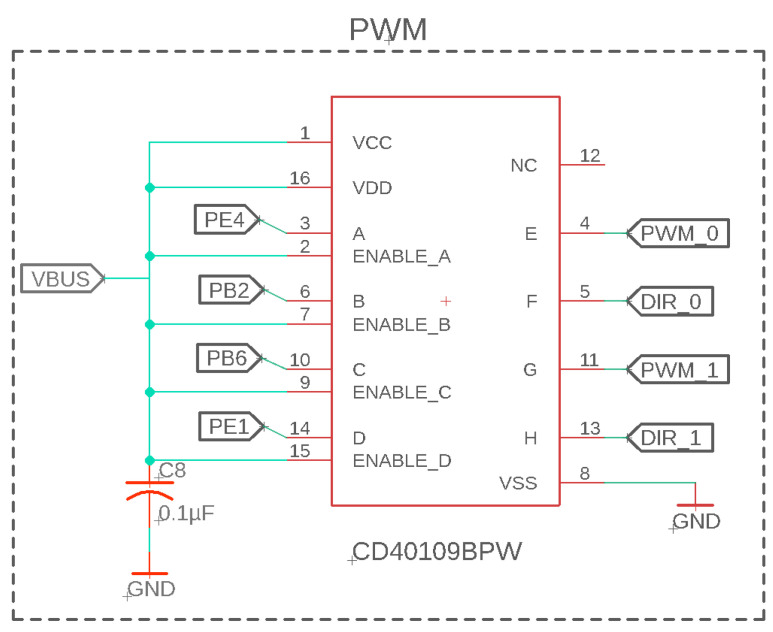
Level shifting from 3.3 V to 5 V logic of the PWM and direction signals for the motor controller input.

**Figure 6 sensors-23-01014-f006:**
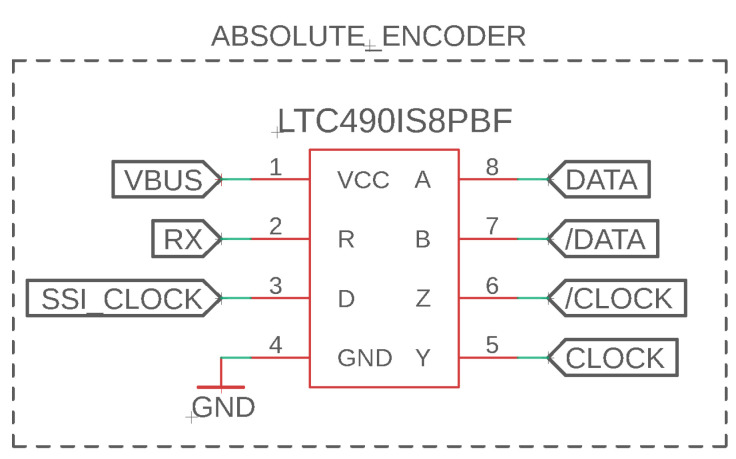
Circuit design for absolute encoders that output differential clock and data signals, which can be interpreted using an SSI module.

**Figure 7 sensors-23-01014-f007:**
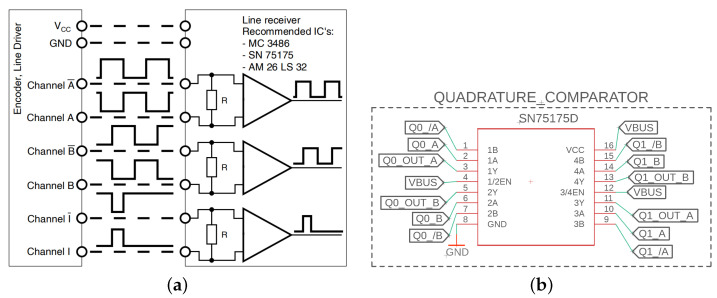
Quadrature encoder circuit utilizes a line receiver that, in practice, is necessary for maintaining channel output integrity to correctly interpret the motor position. (**a**) Line receiver utilized for redundancy (from Maxon catalog [50]). (**b**) Circuit design for quadrature encoders.

**Figure 8 sensors-23-01014-f008:**
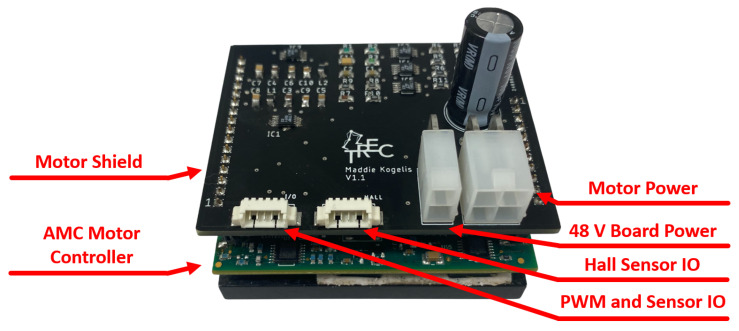
The motor control unit is a combination of an off-the-shelf AMC AZBDC12A8 motor controller and the in-house-designed motor shield.

**Figure 9 sensors-23-01014-f009:**
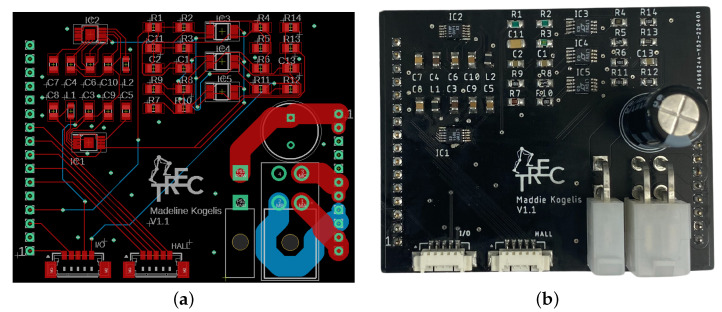
The motor shield is designed to handle power distribution, sensor and motor command routing, and motor phase powering. (**a**) Layout of motor shield PCB. (**b**) Physical build of the motor shield.

**Figure 10 sensors-23-01014-f010:**
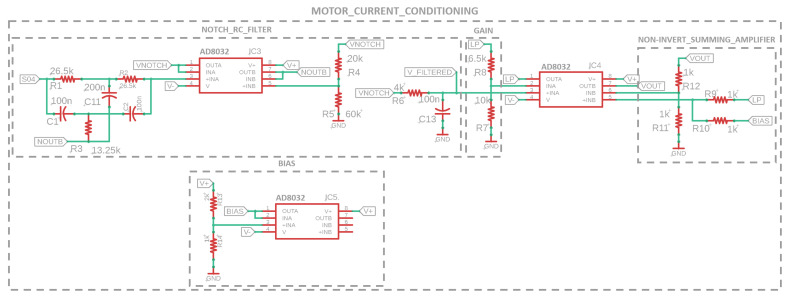
Motor current feedback circuit is made up of a two-stage filter design, a biasing circuit, and a summing amplifier.

**Figure 11 sensors-23-01014-f011:**
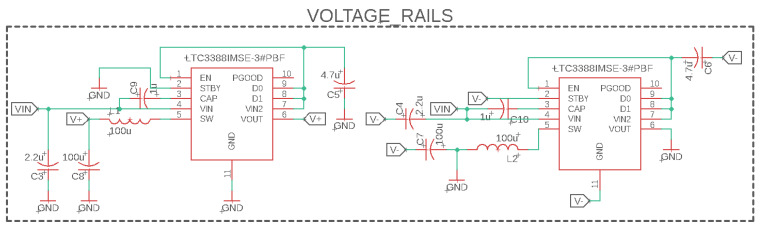
Voltage rail circuit used to achieve ± 5 V ($V_{+}$ and $V_{-}$) output using 5 V input, $V_{IN}$.

**Figure 12 sensors-23-01014-f012:**
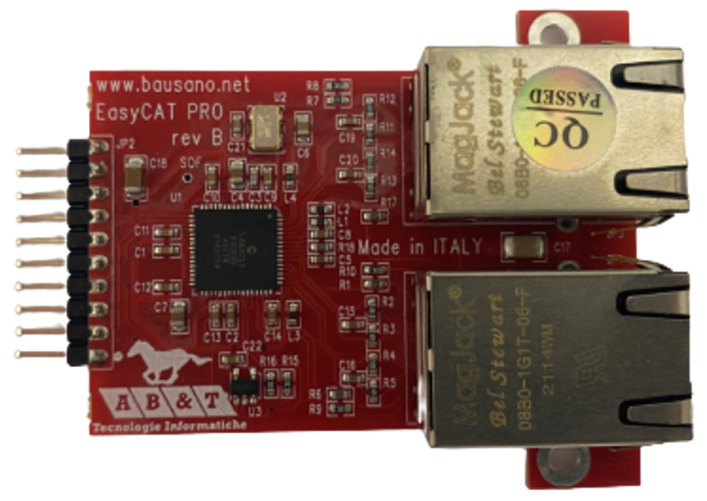
EasyCAT PRO Shield allows for EtherCAT networking capability between the high and low-level controllers.

**Figure 13 sensors-23-01014-f013:**
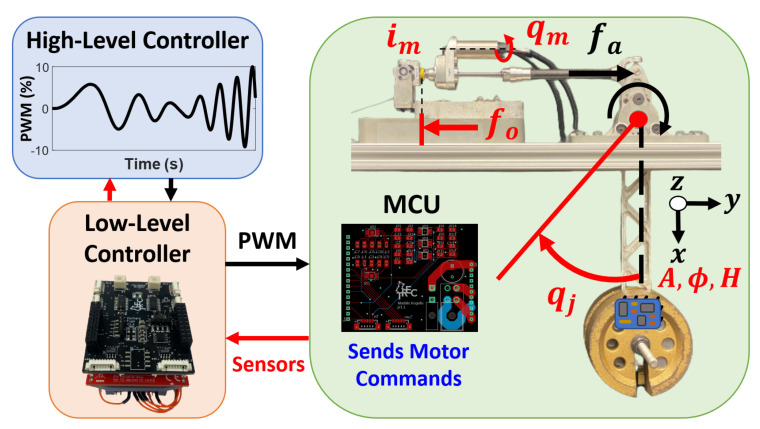
The pendulum test bed has a single LSEA actuator driving a 10 kg mass at the end of a 0.3 m long lever arm with all sensor feedback attached.

**Figure 14 sensors-23-01014-f014:**
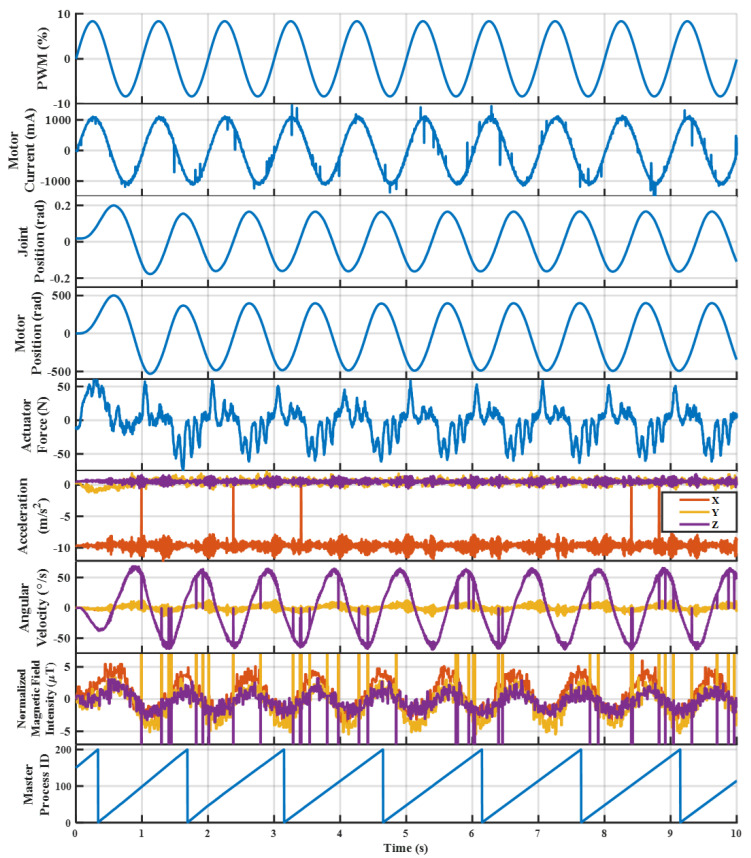
Hardware performance for an open−loop 1 Hz, 1000 mA amplitude sinusoidal input on the pendulum test stand.

**Figure 15 sensors-23-01014-f015:**
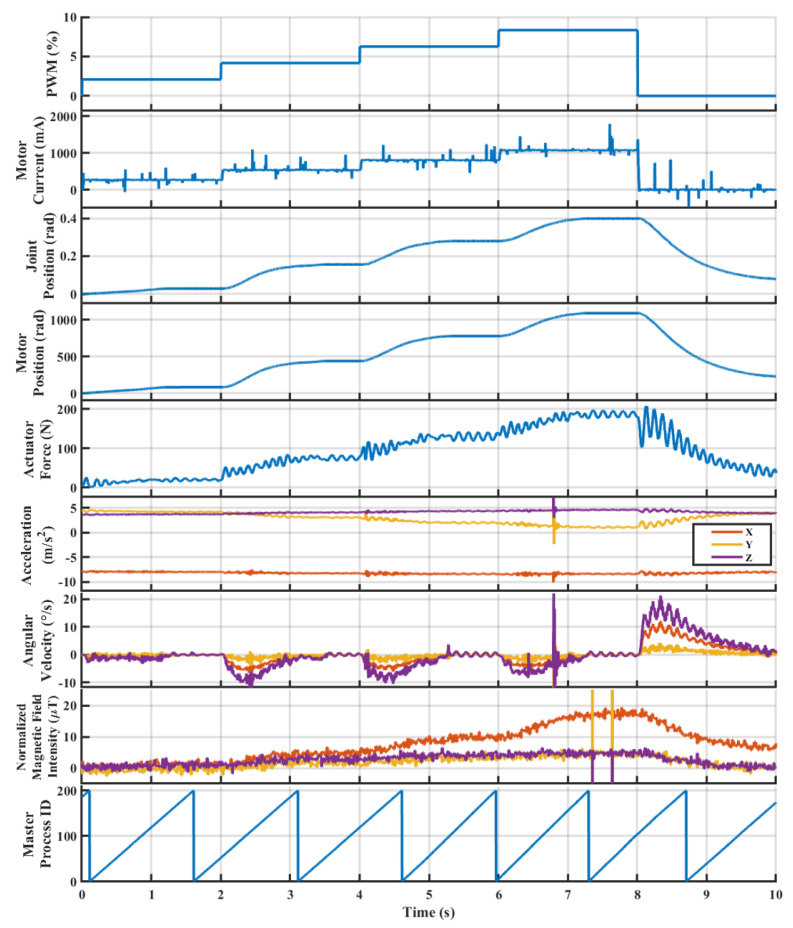
Hardware performance for an open−loop step−wise input of 250 mA, 500 mA, 750 mA, and 1000 mA amplitude on the pendulum test stand.

**Table 1 sensors-23-01014-t001:** CONTROL signal data TIVA-to-master.

Data Type	Data Element	Byte Size
uint8_t	Signal To Master	1
uint8_t	Master Process ID	1
float[]	Force (Actuator 0 and 1)	8 (4 each)
uint32_t[]	Motor Encoder Raw Position (Actuator 0 and 1)	8 (4 each)
int8_t[]	Motor Encoder Direction (Actuator 0 and 1)	2 (1 each)
int32_t[]	Motor Encoder Raw Velocity (Actuator 0 and 1)	8 (4 each)
float[]	Motor Current (Actuator 0 and 1)	8 (4 each)
float[]	Joint Angle (0 and 1)	8 (4 each)
float[]	Accelerometer (X, Y, and Z)	12 (4 each)
float[]	Gyroscope (X, Y, and Z)	12 (4 each)
float[]	Magnetometer (X, Y, and Z)	12 (4 each)
float[]	Force–Torque Force (X, Y, and Z)	12 (4 each)
float[]	Force–Torque Torque (X, Y, and Z)	12 (4 each)
uint8_t[]	Remaining Bytes	24

**Table 2 sensors-23-01014-t002:** CONTROL signal data master-to-TIVA.

Data Type	Data Element	Byte Size
uint8_t	Signal From Master	1
uint8_t	Master Process ID	1
uint8_t[]	Direction (Actuator 0 and 1)	2 (1 each)
float[]	PWM (Actuator 0 and 1)	8 (4 each)
uint8_t[]	Remaining Bytes	116

## Data Availability

Not applicable.

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
