# Peer review of "Design and Validation of a Low-Level Controller for Hierarchically Controlled Exoskeletons"

_sensors, 2023, doi:10.3390/s23021014_

Round 1
Reviewer 1 Report
Great work!
Author Response
We thank the reviewer for going through the manuscript and finding it meritorious.
Reviewer 2 Report
This article presents an implementation of a Low Level controller for a dc motor. Especially, the actuator unit chosen is a SEA. The author describes throughly all the technical aspects of the implementation and the rationale that supports the chosen steps and decisions.In the end, there is a very detailed testing report on the performance of the chosen control architecture.
In my opinion, the main strenght of this paper is that any reader can build, replicate the proposed control architecture and compare the detailed results. This is one of the two main characteristic of a scientific paper, that often is not given for granted.
However, apart for my professional interest in the topic, there is no clear scientific relevant findings shown. At a certain point the paper looks like more a report. In my opinion, it lacks a comparison part with state of the art devices. Or, at least, the main differences that drives the development of the proposed architecture with respect off the shelf alternatives.
In this way, any interested reader, could benefit from this analysis and the paper will help the adoption of this robust and modular architecture.
In fact, there is no State of the Art section, or paragraph, that strongly support the need of such an architecture, there is the target (exos and humanoids) that lacks a proper control architecture, but there is no description apart a few lines at the last paragraph of page 2.
Apart from that, another flaw that I found relevant, is that the PWM in the experiments is created by the High/Medium level controller on the pc rather than in the Low Level control board. Even if the author stated that in the final implementation the High Level controller will set just a torque setpoint.
I think that it could be worthy to show results with the final implementation, mainly because the fact that the major issues with "generic " controllers are related to lag with control loops due to other high computational processes in the pc. In the actual architecture, there could be such problems because of the PWM generated in the High Level control running on a pc.
Lastly, I would suggest to rearrange the Introduction. It is very particular on the exoskeletons that uses SEAs but it is not clearly stated why a partitioned control architecture would perform better. My advice is to reduce slightly on the description of the exoskeletons and focus more on the use of SEA and its advantages. Then split into a subsection, the introduction of the proposed architecture and why it is better with respect to off the shelf components (costs reduction or open access design would be valid metrics).
A few remarks on other parts: there are little errors (like vise-versa that is mispelled) and i strongly suggest to reduce or eliminate the tables reporting data payload. If the author wishes to maintain a graphical representation of the payload messagge, please use a standard representation like the one attached, it will be clearer and use less space.

Author Response
Please refer to the provided document for the full response.

Reviewer 3 Report
The paper deeply details the application and the technical solution of a low level controller in a exoskeleton. It is well written and covers the design and validation process of the controller.
However, this is also the main problem. There is too much focus on the technical solution, turning the paper in a technical report. The scientific contribution of the paper to the field is not clear.
My first suggestion is to make the contribution clearer and tie it with the respective references, to show the novelty of the problem/solution.
Moreover, this alteration probably will need some changes in the paper structure to better focus on this contribution, especially on the methods and results.
The second main problem is the references choice. There are at least 10 self-citations, and they should be greatly reduced or completely removed. Moreover, more updated references from the past 5 years will improve the paper quality.
Detected self citations: Herron [4, 18, 37, 42, 43], Kalita [5, 11, 37, 42, 43], Leonessa [4, 18, 27, 35, 37, 42, 43]
At the moment, I can't recommend the paper because of the mentioned key points. However, the paper topic is very interesting and it has potential for publication.
Author Response
Please refer to the attached document for the full response.

Reviewer 4 Report
This manuscript serves as a very thorough exploration of a low-level control system created by the authors. I could see this serving as a good reference manuscript for future work.
I believe that this manuscript is good, but have a few minor criticisms that should be resolved before publication:
1. There are some grammatical issues throughout - these are minor, but persistent throughout the document. This could be remediated with another read-through.
2. Figure 1, while useful, is quite messy - particularly the LSEA diagram. With this figure being referenced quite a bit throughout the introduction, it may serve the authors well to clean up the diagram. Perhaps even having the more complex LSEA diagram as a stand-alone graphic?
3. In 2.3, you discuss rejecting bending loads, but don't provide any support or rationale. I understand why you are doing this, but justify your decision to do-so.
4. For your citation of [46], it seems that you address an individual by name, "originally proposed in Shriyah." It's a bit unclear how this is worded and perhaps could be reworked to appropriately address that author's work.
5. In 2.4, you discuss that the PWM and direction logic requires 5V. This is standard for many h-bridge type drivers. I don't think it is required to discuss that you found this out via the manufacturer of your particular driver.
6. While I agree that the combination of absolute/incremental encoders with a Kalman filter is beneficial, you should cite evidence of this being a "greater approach." There may be some of your readership that would appreciate clearer justification of this methodology.
7. Was drift ever considered as an issue with the MPU9250 or was it ever explicitly accounted for? I know that vibrations can induce drift - requiring some form of compensation, I just don't know if there would be any significant effects with this experimental setup.
8. I am not entirely convinced that the transfer function presented in (4) adequately represents all of the filtering going on. First of all, it doesn't seem like the time-constant term (RC) has been manipulated properly in the transfer function (as written, the RC term would not be properly associated with the 1st order term). Check to make sure that this is accurately representing a low pass filter. Second, perhaps your notch-filter is worthwhile to include in this? If this equation is merely serving as a means of parameterizing your R and C values, maybe this could just be more concisely presented?
9. Across equations 5 - 8, the redundant use of V_out is unclear. Similarly, the introduction of the term V_LP in the last sentence is a bit unclear.
10. I'm a bit confused in table 1. Why are utilizing a 32-bit integer for the encoder direction when it appears to utilize a single byte?
11. In Figure 11, there are a great deal of variables, some of which are not well-described in-text. The graphic title could better establish all of these parameters.
Author Response

(The authors gave the same response as above.)

Round 2
Reviewer 2 Report
I have read the replies to my comment and wanted to thank the authors for their fast, punctual and precise answers. In addition, I want to thank for understanding my points to strenghten this manuscript.
From the replies and the introduction of new paragraph I think that the manuscript has been patched to overcome the major flaws that I found in the first draft.
To conclude I really appreciate the fact that tha authors decided to release the project as open-access (edited with a freeware software, too). This is truly adherent to the very aim of the manuscript, overcoming the problem (that in my opinion has been summarized in an excellent way) as :"development of the same hardware is unnecessarily time-consuming, expensive, and outside the scope of the core research objectives" .
Author Response
Please see the attached document for the full response.

Reviewer 3 Report
The authors did improve the paper and added the scientific contribution of the paper, although, some of the contributions are more technical than scientific. However, it is ok. The paper quality and its details compensate this aspect.
On the other side, even though the authors explained the use of self citations, in my opinion, self citations really should be used as a last resort option, when a explanation or validation is really necessary and there is no other work to sustain it.
I do agree that some of the self references are essential, but there a few of them can be removed. I suggest to remove at least these [4, 5,11, 35], as they do not have a heavy impact on the paper. I'm not sure if there is a magic self citation number for Sensors approval, thus I will leave this aspect for the Editors.
Author Response

(The authors gave the same response as above.)
